# Position: Beyond the Sequence – Graph Learning as the Blueprint for Trustworthy Large Language Models

## Abstract

A common working assumption in modern AI says that scaling generic architectures on massive datasets, principally Large Language Models, suffices for general intelligence, rendering explicit inductive biases such as those offered by Graph Learning (GL) unnecessary. In this position paper, we challenge the exclusivity of this view, arguing instead that GL serves as a structural multiplier to the statistical power of scaling. We identify a core representational mismatch: recovering relational topology from serialized sequences necessitates increasingly resource-intensive scaling, leaving persistent deficits in factual grounding, logic, and transparency that agnostic scaling alone cannot economically resolve. We contend that GL bridges this gap by explicitly modeling relational dependencies, providing the structural scaffolding for efficient and verifiable reasoning. We operationalize this position through four strategic integrations: (i) Context Engineering, extending flat buffers with graph-structured memory; (ii) Architecture, recasting self-attention as learnable token-interaction graphs; (iii) Interpretability, mapping neural dynamics onto discrete causal subgraphs; and (iv) Safety, shifting from behavioral filters to topological governance. Ultimately, we advocate that integrating graph-based priors does not constrain the model, but rather disciplines the raw power of scale, ensuring that the pursuit of general intelligence remains both verifiable and economically sustainable.

## 1. Introduction

The success of Large Language Models (LLMs) is founded on the "scale-first" hypothesis: the premise that general intelligence emerges naturally from massive increases in parameters, data, and compute (Kaplan et al., 2020; Hoffmann et al., 2022). While this paradigm has enabled models to capture profound statistical patterns in language with unprecedented fidelity, it faces inherent bottlenecks rooted in its fundamental data representation. Current LLMs attempt to recover the real world's multi-dimensional, non-linear topology of relational dependencies from one-dimensional sequences of tokens via resource-intensive scaling. This process leaves persistent deficits in factual grounding, logic, and transparency that agnostic scaling alone may not economically resolve.

By serializing inherently relational data, LLMs tend to prioritize local token transitions over global consistency, often compromising factual grounding (Ji et al., 2023; Huang et al., 2025). This also renders the model's learned knowledge rigid and surgically inaccessible; because knowledge is dissipated across dense, unstructured weights, the model lacks the explicit relational anchors required for targeted updates or precise knowledge updating (Dhingra et al., 2022a). Finally, the reliance on an unstructured latent space limits mechanistic interpretability (Geva et al., 2023). Internal reasoning remains largely opaque, where model explanations often function as post-hoc rationalizations rather than reflections of a verifiable logical process (Turpin et al., 2023).

**Our Position:** We argue that advancing trustworthy LLMs may benefit critically from an explicit relational substrate. We posit that Graph Learning (GL) can provide this framework as a scaling multiplier, recasting relational dependencies from latent artifacts into first-class computational objects. This shift is particularly vital for domains governed by formal rules, such as knowledge hierarchies, task workflows, or the physical symmetries of molecules (Bronstein et al., 2021). The efficacy of GL stems from its capacity to codify these structural inductive biases, offering a more sample-efficient and reliable path to inference than the agnostic pattern recognition of sequence-only models. For instance, subgraph reasoning naturally encapsulates the mechanics of multi-step logic (Teru et al., 2020; Wang et al., 2025c), while Graph Neural Networks (GNNs) achieve algorithmic alignment by mirroring the discrete execution of logical and combinatorial operations (Cappart et al., 2023; Mor-

[1]Anonymous Institution, Anonymous City, Anonymous Region, Anonymous Country. Correspondence to: Anonymous Author <anon.email@domain.com>.

Preliminary work. Under review by the International Conference on Machine Learning (ICML). Do not distribute.

ris et al., 2023). While the "Bitter Lesson" (Sutton, 2019) suggests that resource-intensive scaling might eventually recover these structures, the reality of looming data exhaustion (Villalobos et al., 2024) and the diminishing marginal utility of unstructured data scaling (Chen et al., 2025) suggests a practical ceiling. We contend that the relational perspective of GL is the critical mechanism for steering the raw power of scale toward grounded, verifiable reasoning.

We identify several key areas where GL can serve as a structural blueprint for LLMs, built upon four foundational paradigms. First, topological centrality utilizes network connectivity and diffusion mechanisms to quantify information importance, providing a mathematical framework to filter noisy contexts by prioritizing data based on structural authority rather than mere semantic similarity (Page et al., 1999b). Second, structural message passing empowers neural architectures to enforce strict inductive biases and permutation symmetry—fundamental invariance properties that will be compromised when complex relational data is linearized into sequences (Kipf & Welling, 2017; Hamilton et al., 2017). Third, graph embeddings and geometric regularity provide a basis for decoding internal representations by mapping neural dynamics into continuous vector spaces that preserve essential topological properties (Belkin & Niyogi, 2003); this aims to ground interpretability in the intrinsic manifold structure of latent features rather than superficial linguistic patterns. Finally, relational knowledge modeling offers the formal machinery to transform dense, heuristic representations into interpretable reasoning chains and supporting subgraphs (Schlichtkrull et al., 2018; Ji et al., 2021), establishing a foundation for topological defense against structural safety vulnerabilities.

Translating these paradigms into actionable research frontiers, we propose four critical blueprints that redefine the structural integration of GL within the LLM lifecycle.

**A Context Engineering Blueprint: From Flat Retrieval to Structured Memory.** While standard Retrieval-Augmented Generation (RAG) reduces hallucinations, it often struggles with multi-hop queries that require traversing complex relational dependencies (Yang et al., 2018a; Google, 2025). We advocate augmenting flat context buffers with graph-structured abstractions to better exploit available structural priors. By prioritizing structurally authoritative information, graph-based retrieval denoises context, effectively filtering out distractors that are semantically similar but logically irrelevant (Gutiérrez et al., 2024; Li et al., 2025b). This approach also extends to Agentic Context Engineering, where evolving graph structures model logical and temporal dependencies, supporting agentic workflow synthesis (Li et al., 2025a) and dynamic episodic memory for coordination (Chhikara et al., 2025; Sun et al., 2025).

**An Architectural Blueprint: Reforming the Attention Bottleneck.** This blueprint proposes that properly structuring dense self-attention can help alleviate the quadratic complexity and memory bottlenecks inherent in long-context LLMs. By treating self-attention as a learnable token interaction graph, structural priors can be leveraged to optimize KV-cache management through relational eviction and importance-aware compression (Zhang et al., 2023; Li et al., 2025d). Beyond efficiency, this framework moves toward graph-native attention, where token interactions are modeled as message passing over directed acyclic graphs that better approximate logical flow. By integrating structural information into the models' internal computation, we aim to preserve essential structural inductive biases and eliminate the positional biases and signal dilution caused by naive serialization (Wang et al., 2025a; Liu et al., 2025).

**An Interpretability Blueprint: From Black-Box Heuristics to Mechanistic Clarity.** This blueprint proposes leveraging graph-based abstractions to better penetrate the black box of LLM reasoning. Instead of relying solely on post-hoc verbal rationalizations, which can be unfaithful reflections of the model's true decision-making (Turpin et al., 2023; Lanham et al., 2023), we advocate mapping abstract neural dynamics onto discrete, interpretable structures. By identifying sparse component circuits and projecting activations onto semantic implicit graphs (Lindsey et al., 2025b), this framework attempts to decode how specific concepts propagate across neural layers. This structural clarity transforms interpretability from passive observation into a more verifiable, actionable process, providing a principled surface for causal intervention and model regulation.

**A Safety Blueprint: From Atomic Erasure to Topological Defense.** This blueprint argues that robust AI safety requires treating an LLM's internal knowledge as an interconnected relational system, not a collection of isolated facts. Many current approaches succumb to the "Atomic Fallacy," leaning on superficial prompt-level filters that disregard the model's dense associative structure. We contend that reliability instead demands protecting the model's topological integrity, promoting consistency in internal pathways and causal connections. Under this view, effective unlearning aims to make information topologically unreachable in the model's implicit knowledge graph rather than merely obfuscated (Wu et al., 2024a). The same graph-centric lens also supports the detection of adversarial incursions by flagging anomalous activation trajectories as they propagate through relational pathways (Wei et al., 2025b). Ultimately, the blueprint shifts safety from reactive behavioral defenses toward verifiable, structural governance of internal logic.

## 2. Graph Learning as the Blueprint for LLMs

In this section, we detail the above four technical blueprints, which constitute a cohesive lifecycle approach for LLMs,

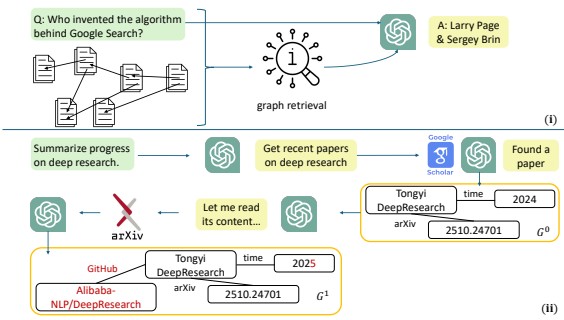

*Figure 1.* **Illustration of graph-based context engineering. (a)** RAG with a static graph-based corpus; **(b)** Evolving graph for agentic memory.

spanning context engineering, architecture, interpretability, and safety.

### 2.1. Graph-based Context Engineering

The reliability of LLMs is frequently undermined by factual hallucinations, knowledge staleness, and performance degradation in long-context scenarios. These limitations have catalyzed a shift toward context engineering (Mei et al., 2025) which emphasizes the systematic curation of input to ground model generation. Although Retrieval-Augmented Generation (RAG) (Lewis et al., 2020) is the prevailing standard for such groundings (Shuster et al., 2021; Dhingra et al., 2022b; Xu et al., 2024), most implementations are constrained by treating context as a flat, disconnected buffer of text, thereby neglecting inherent structural dependencies. While recent works have begun exploring structured context (Gutiérrez et al., 2024), these efforts remain specialized, and the community remains skeptical of whether the benefits of graph-structured representations outweigh the costs of graph construction (Edge et al., 2024). We propose that augmenting context engineering with graph-structured abstractions, particularly where structural priors are available, offers a distinct advantage. The potential benefits of this shift range from enhanced information management in RAG to robust state-tracking for long-horizon agentic reasoning.

**Established Advantages: Signal Discovery via Graph-Structured Context (Fig. 1 (a)).** Despite the rise of long-context models (Yang et al., 2025a; Li et al., 2024a), the prohibitive computational cost of massive buffers renders retrieval indispensable. Here, the utility of graph traversal for multi-hop reasoning has been increasingly validated in the LLM era (He et al., 2024; Mavromatis & Karypis, 2025; Luo et al., 2025b; Edge et al., 2024). While Dense Retrieval (Karpukhin et al., 2020) and BM25 (Robertson et al., 1994) struggle to bridge documents that are logically related but semantically distinct (Yang et al., 2018a), graph-based approaches resolve this by traversing explicit links rather than relying on vector similarity. As confirmed by recent studies (Han et al., 2025; Dong et al., 2024a; Xiang

*Table 1.* Graph RAG improves retrieval and suppresses harmful distractors on the graph-based long context HaystackCraft dataset (Li et al., 2025b) which targets solve QA tasks using the full English Wikipedia network (7M articles, 100M edges, 30B+ tokens in total) as the corpus. In this setup, BM25 serves as the base retriever, with PPR applied for reranking.

*(a).* Retrieval recall@$K$ improvements from reranking.

| | | | $K$ | |
|---|---|---|---|---|
| PPR | 20 | 40 | 80 | 160 |
| | 0.428 | 0.474 | 0.531 | 0.587 |
| ✓ | **0.550** | **0.600** | **0.636** | **0.666** |

*(b).* Under perfect retrieval coverage, PPR mitigates harmful distractors and leads to improved long-context question answering performance in F1 score (↑).

| Model | PPR | # Context Tokens ($\times 10^3$) | | |
|---|---|---|---|---|
| | | 32 | 64 | 128 |
| Llama-3.1-8B-Instruct | | 46.16 | 42.06 | 37.24 |
| | ✓ | **49.81** | **45.87** | **42.80** |
| GPT-4.1 mini | | 64.28 | 60.73 | 60.55 |
| | ✓ | **64.46** | **63.36** | **62.20** |

et al., 2025) and our results on the HaystackCraft benchmark (Tab. 1(a)), algorithms such as Personalized PageRank (PPR) significantly outperform semantic baselines in recovering these structural dependencies, translating to correct downstream responses (Li et al., 2025b; Page et al., 1999a).

**The Overlooked Critical Insight: Mitigating Semantic Interference.** A less understood but equally critical advantage of Graph-based RAG is its role in noise mitigation during context preparation. In practice, retrieved context inevitably contains distractors. Standard retrievers, which prioritize maximizing semantic similarity, often fetch semantic distractors that sound highly relevant to the query but are factually unrelated. This is particularly hazardous as modern LLMs operate fundamentally as semantic attention engines; when conditioned on high-similarity noise, they are prone to confusion, preferentially attending to plausible-sounding but incorrect tokens over the ground truth.

Graph retrievers, by contrast, operate on a different signal: topological connectivity. This allows the system to balance semantic evidence with structural proximity. When graph retrieval introduces noise (e.g., irrelevant neighbor nodes), this noise is typically less semantically similar to the query than that of dense retrievers. We characterize these as weaker distractors that are distinct enough for the LLM to easily filter out. Therefore, Graph-based RAG serves as a logic-based ensemble to the LLM. By diversifying the retrieval logic beyond pure language modeling, it can improve the signal-to-noise ratio of the context window. As shown in Tab. 1(b), this difference is empirically significant: LLMs grounded by graph-retrieved context consistently

yield higher accuracy than those using standard retrieval, particularly in long-context settings where robustness to distractors is essential.

**Graphs for Dynamic State Tracking in Agentic Memory.** While RAG addresses static factual grounding, autonomous agents in workflows such as Deep Research (Google, 2025) and Automated Software Engineering (Jimenez et al., 2024) introduce a harder challenge: dynamic state evolution. Current architectures primarily rely on linear context buffers, i.e., flat logs of (observation, action) pairs, but this abstraction is ill-suited for complex, multi-step tasks because it conflates temporal sequence with logical dependency. As trajectories lengthen, retrieving specific past decisions forces the agent to scan unstructured history, exacerbating the "lost-in-the-middle" phenomenon (Liu et al., 2024).

We advocate using evolving graphs as the architectural standard for agentic memory. Instead of merely appending tokens to a buffer, agents can dynamically maintain a graph-structured memory where nodes encapsulate entities, episodic states, or interaction summaries, and edges encode relational, causal, and hierarchical dependencies. For instance, a deep research agent can continuously refine its graph memory as it synthesizes dynamic information from Google Scholar, arXiv, and GitHub (Fig. 1 (b)). This paradigm shift is empirically supported by frameworks such as Mem0 (Chhikara et al., 2025), A-Mem (Xu et al., 2025), and Zep (Rasmussen et al., 2025), which utilize graph structures to distill persistent entity-centric states from transient conversation logs. This enables structured introspection, allowing agents to debug errors by traversing specific causal links rather than scanning unrelated temporal events.

Furthermore, graph memory can evolve the Context Folding mechanism (Sun et al., 2025; Ye et al., 2025) into a lossless, navigable structure. While existing folding methods effectively solve context clutter by collapsing resolved subtasks into summaries, they often permanently discard intermediate states. In contrast, graph memory may fold these subtasks into single supernodes while preserving pointers to underlying details. This specifically allows historical specifics to be unfolded on demand, offering a dynamic resolution capability that is impossible in linear buffers.

**Call to Action: End-to-End Optimization of Graph Topologies.** Despite the demonstrable utility of graph-structured context, its adoption is bottlenecked by the rigidity of graph construction. Current paradigms typically rely on open information extraction, implemented either via static models (Angeli et al., 2015) or LLM-based prompting (Edge et al., 2024). However, these approaches often impose a fixed granularity of abstract concepts that may be suboptimal for reasoning. While semi-structured graphs (where nodes represent text segments rather than abstract concepts) offer flexibility (Gutiérrez et al., 2024; Wu et al., 2024b),

some misalignments persist: powerful LLMs benefit more from coarse-grained semantic blocks, whereas weaker models often require fine-grained entities to bridge reasoning gaps (Li et al., 2025c). We argue that optimal granularity is not universal, but should be adaptive to the LLM's capability. Current workflows prioritize human semantic intuition on graph construction, often failing to maximize the agent's specific reasoning potential.

To address this gap, we suggest exploring a transition from manual graph engineering to end-to-end structural optimization. While recent advances show the potential of teaching LLMs to manage memory via reinforcement learning (Yu et al., 2025; Zhou et al., 2025b; Sun et al., 2025), current approaches, such as Graph-R1 (Luo et al., 2025a), apply RL to graph retrieval while still retaining manually specified schemas and prompt-based construction. This leaves graph construction disconnected from the task reward. We propose that graph construction could be optimized as an intrinsic component of the reasoning trace. By atomizing operations, such as parsing text, folding nodes, or linking states, into optimizable actions, agents can learn to build structures dynamically. While the resulting search space is vast, it can be tractably constrained by heuristic bounds and guided by fine-grained process rewards, akin to tool usage learning. We hypothesize that agentic performance will ultimately prove more robust to variations in process reward design than to the brittleness of fixed graph schemas.

### 2.2. Imposing Graph Priors on the Architectures

The reliance of current LLMs on full causal self-attention represents a brute-force approximation of dependency modeling: it assumes dense connectivity among all tokens to eventually recover a sparse set of meaningful relationships. This inefficiency becomes untenable as context lengths expand, motivating the adoption of KV-cache compression and sparse attention mechanisms to better manage the context (Jiang et al., 2024; Tang et al., 2024). Previous findings in KV-eviction confirm that intrinsic information flow is highly sparse, with models retaining under 20% of their cache without degradation (Zhang et al., 2023; Ge et al., 2024; Cai et al., 2024; Kim et al., 2025). However, existing methods typically exploit this sparsity via post-hoc heuristics, reconstructing latent dependencies on the fly using proxies, such as accumulating attention scores (Zhang et al., 2023; Li et al., 2024b) or hash-based retrieval (Kitaev et al., 2020; Mu et al., 2025), rather than leveraging available prior knowledge regarding the structure within the data.

Flattening inherently structured data, such as reasoning chains, causal graphs, or code dependency trees, into a serialized stream dilutes the topological signal, requiring models to expend capacity rediscovering dependencies that were already explicit in the source (Wang et al., 2024a). This

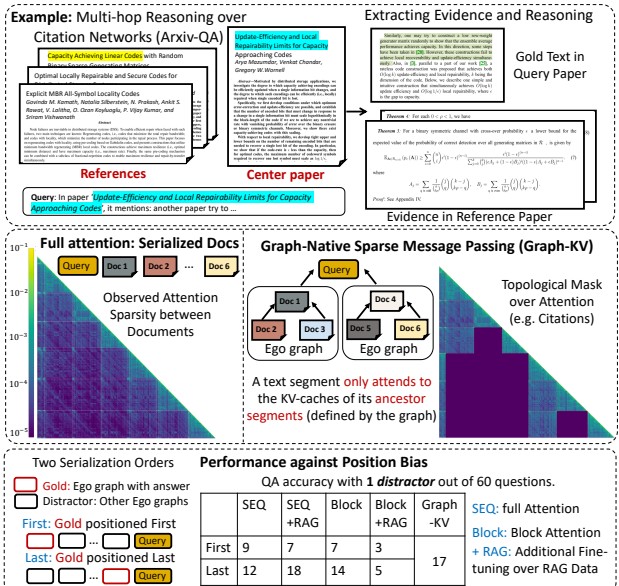

*Figure 2.* **Illustration of Cross-Block Sparsity on the Arxiv-QA benchmark** (Wang et al., 2025a). **Top:** A multi-hop QA example requiring the model to reason over a center paper and its cited neighbors (ego-graph) to extract evidence. **Middle:** Comparison of attention density. While Full Attention naturally exhibits soft sparsity matching the citation structure, Graph-KV enforces a strict topological mask (DAG-masked attention), restricting attention solely to valid ancestor segments. **Bottom:** Robustness against position bias. We evaluate performance when the gold ego-graph containing the answer is positioned first versus last in the context. Baselines exhibit significant variance due to position bias, whereas Graph-KV maintains consistent high accuracy regardless of serialization order. All models are variants of Llama 3.1-8B.

misalignment introduces positional biases that can override logical relevance, often hindering the extraction of even shallow logical chains from long contexts (Liu et al., 2024; Wu et al., 2025). To address this, we advocate for integrating LLMs with the inductive biases of graph-aligned architectures (e.g., GNNs and Graph Transformers); these models naturally enforce permutation invariance and preserve structural priors conducive to robust reasoning.

**Independent Sets and Block Attentions.** The transition away from the dense sequence assumption is already evident in frameworks including PCW (Ratner et al., 2023), APE (Yang et al., 2025b), and Block-Attention (Ma et al., 2024). These methods operate on the premise that computing cross-attention among retrieved documents is often computationally unnecessary. We posit that they represent a simplified instantiation of our graph blueprint: by modeling context as disconnected components ($\mathcal{G} = (\mathcal{V}, \emptyset)$) and restricting attention flow to occur only between the query and individual nodes, they reduce complexity from quadratic to linear. This structural perspective is empirically validated by BlockRank (Gupta et al., 2025), which leverages such inter-document attention sparsity to achieve superior effi-

ciency over naive sequential concatenation for generative reranking tasks.

**Generalizing to Directed Acyclic Graphs (DAGs).** While independent sets address efficiency, complex reasoning requires capturing the logical flow between nodes. Therefore, we advocate for generalizing this paradigm to arbitrary DAGs. This approach reinterprets the Transformer (Vaswani et al., 2017) through the lens of graph attention (Veličković et al., 2018), treating it as a differentiable engine for DAG traversal. Graph-KV (Wang et al., 2025a) extends the parallel encoding concept by re-introducing edges, targeting generative QA tasks. Given a Structural Context $\mathcal{G} = (\mathcal{V}, \mathcal{E})$ capturing dependencies between text blocks (e.g., citations, function calls), it enforces a topological mask where a node $v$ attends strictly to its ancestors $\{u \mid \exists \text{ path } u \rightarrow v \in \mathcal{G}\}$. Crucially, to resolve the positional bias inherent in serialization, it employs Graph-aligned Positional Encoding, assigning shared PE ranges to logically parallel nodes in the DAG. On the ARXIV-QA benchmark involving multi-hop reasoning over citation networks, Graph-KV reduces computational costs via DAG-restricted attention and maintains robust accuracy across shuffled contexts, whereas standard sequential models suffer significant performance degradation due to positional biases (see the validation in Fig. 2).

This topological view is equally critical for representation learning. Struc-Emb (Liu et al., 2025) demonstrates that by treating related documents (via citations or hyperlinks) as parallel nodes via cached KV states, the model avoids the signal dilution caused by flattening. Extensive zero-shot experiments on structure-rich benchmarks such as MuSiQue (Trivedi et al., 2022) and HotpotQA (Yang et al., 2018b) show that this in-process, structure-aware encoding consistently outperforms text-only baselines. Crucially, comparisons between structural sparse encoding (attending only to relevant neighbors) and full sequential embedding (concatenating all context) reveal that the sparse approach scales robustly to long contexts, whereas dense full embedding collapses due to the "needle-in-a-haystack" effect.

These findings indicate that graph priors provide not only computational efficiency but also robust inductive biases for encoding data topologies. However, while this paradigm shares core principles of structural sparsity and permutation invariance with GNNs, it necessitates distinct adaptations for generative tasks. Unlike traditional GNNs, which typically employ bidirectional message passing over vector embeddings, graph-guided LLM architectures operate on coarse-grained semantic blocks, such as paragraphs, docs, or reasoning chains, and strictly adhere to causal, decoder-only directionality. This distinction enables the model to preserve deep semantic coherence within local blocks while aligning global information flow with the logical dependencies essential for valid reasoning and code execution.

**Call to Action: Towards Structure-Aware Architectures.** Despite early successes, realizing the full potential of structure-aware LLMs requires concerted research efforts to address the resulting distribution shifts. Current methods often impose graph masks on off-the-shelf LLMs at inference time; however, pre-trained attention heads are rarely optimized for the abrupt sparsity induced by such topological masking. While frameworks such as PCW (Ratner et al., 2023) and BlockRank (Gupta et al., 2025) successfully utilize fine-tuning for simplified set structures, we advocate for an extensive investigation of alignment tuning strategies that adapt models to more complex structural priors in the data. Furthermore, we posit that Agentic Workflows represent the ideal testbed for this paradigm shift. Agentic reasoning naturally manifests as a dynamic, evolving DAG of tool usage and state dependencies. Emerging systems such as Tongyi's Deep Research (Li et al., 2025a) exemplify the potential for high-quality structural agentic flow synthesis. Moreover, to ensure robustness against noisy real-world topologies, future research may also explore soft structural modulation, where graph priors act as learnable attentional biases rather than rigid hard masks.

### 2.3. Explaining LLMs via Graph Approaches

Integrating LLMs into high-stakes sectors, such as healthcare and law, highlights the need to move beyond simple behavioral observation toward more rigorous verification. Traditional metrics are often insufficient as they fail to capture the causal mechanisms that determine a model's reliability. To improve transparency, we need a format that maps abstract neural dynamics onto discrete, understandable structures. We propose that graph-based modeling holds great potential to bridge this gap. By viewing interpretability through a topological lens, we can more effectively track the flow of information, help identify the root causes of failure, and support efforts toward formal regulation.

**Graph-based Explanations of Internal Mechanisms.** A prevailing assumption in the community is that LLMs are naturally interpretable through their generated text, exemplified by the chain-of-thoughts (Wei et al., 2022). However, we argue that relying solely on surface forms faces inherent limitations. Growing evidence suggests that verbalized explanations often serve as post-hoc rationalizations rather than faithful representations of the model's true decision-making process (Lanham et al., 2023). For instance, Turpin et al. (2023) observed that models typically adopt external biases (e.g., user suggestions) yet generate reasoning chains that avoid acknowledging this influence. Trustworthy explanation, therefore, cannot rely on the surface sequence alone; it requires analyzing the underlying causal structure of the decision process. We suggest that graph-based abstractions are well-suited to resolve the high-dimensional, polysemantic nature of these internal mechanisms.

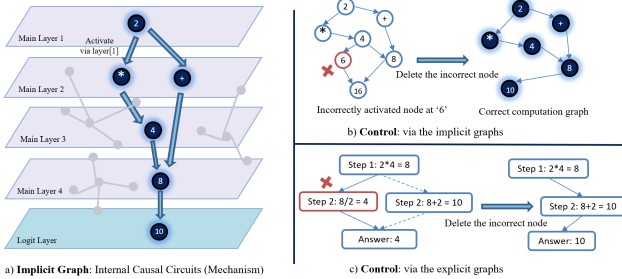

*Figure 3.* **Graph-based explanation and control of LLM reasoning.** Using the example "$2 * 4 + 2$", **(a)** implicit graphs represent internal causal circuits across model layers; **(b)** control via implicit graphs corrects reasoning by intervening on incorrectly activated internal nodes; **(c)** control via explicit graphs prunes invalid surface-level reasoning steps to enforce structural consistency.

Mechanistic interpretability offers the empirical basis for this graph-centric view, although existing techniques are still fragmented. Pioneering frameworks such as ACDC (Conmy et al., 2023) and EAP (Syed et al., 2024) have validated this perspective by uncovering sparse "component circuits" within dense models. However, the resulting structures often lack semantic clarity, as the graph nodes correspond to raw, polysemantic architectural indices (e.g., specific attention heads) rather than human-interpretable concepts. By treating these disentangled features as the true graph nodes, we can recover fine-grained information flow graphs that reveal how specific concepts propagate and interact throughout the network (see Fig. 3 (a)) (Lindsey et al., 2025a; Dai et al., 2025). While these methods show the potential of graph-based interpretation, current approaches are still difficult to scale, posing a challenge for applying this to the largest frontier models.

**Graph-based Control and Regulation of Model Behavior.** Beyond passive explanation, graph-structured abstractions can also be used for active control and regulation of model behaviors. By exposing the causal structure of computation, these frameworks enable precise interventions to rectify errors and enforce safety constraints. For instance, GraphGhost (Dai et al., 2025) demonstrates that targeted interventions on feature-level nodes, such as those identified by SAEs, can causally steer reasoning trajectories (see Fig. 3 (b)). Similarly, Circuit-based Reasoning Verification (CRV) (Zhao et al., 2025c) employs graph-level structural properties to distinguish reliable reasoning from hallucination. Together, these works suggest that graph-based explanations offer a powerful control surface for mechanistic verification, extending significantly beyond the capabilities of black-box methods.

While we previously argued that raw generated text is an unreliable proxy for internal thought, structuring this output into explicit graphs provides a robust signal for behavioral adjustment and alignment. Unlike implicit graphs, which

encounter significant scalability hurdles in semantic extraction, explicit graphs are defined directly over observable tokens and transition states (see Fig. 3 (c)) (Minegishi et al., 2025; Matsutani et al., 2025), making them inherently interpretable and compatible with standard training pipelines. Recent work demonstrates that modeling reasoning trajectories as explicit graphs facilitates the internalization of structured logic, for instance through graph-based data augmentation (Wang et al., 2024b). Furthermore, by reframing the generation of tokens as a directed flow over a reasoning graph, we can enforce graph-theoretic priors, such as optimizing for maximum information flow or structural consensus, directly within the learning objective (Anonymous, 2025). This approach shifts the optimization focus from local token likelihoods to global structural coherence.

**Call for Action: Toward Topological Alignment.** The frameworks discussed above suggest a complementary path: leveraging explicit graphs for scalable modulation while reserving implicit graphs for precise verification. However, fully realizing this potential requires overcoming a critical gap: the misalignment between the explicit control interface (the reasoning graph) and the implicit computational engine (the neural activations). Currently, these two layers are often decoupled. Consequently, we argue that future research may prioritize topological alignment. We envision a paradigm where the Explicit Reasoning Graph serves as a high-level causal abstraction that faithfully constrains the low-level Implicit Computation Graph. By establishing such alignment, we can ensure that the graph-based explanations we see on the surface are not merely post-hoc rationalizations, but faithful representations of the underlying mechanism, enabling trustworthy control and modulation.

### 2.4. LLM Safety: The Structural Imperative

While the previous sections addressed trustworthiness through the lens of reliability and transparency, institutional trust necessitates rigorous knowledge governance, encompassing privacy, copyright compliance, and unlearning (Eldan & Russinovich, 2023; Qiu et al., 2025; Meng et al., 2022a), while safety focuses on enforcing robust controls to enforce policy-compliant responses (Zou et al., 2023; Dong et al., 2024b; Zhang et al., 2024). However, we contend that current approaches suffer from the **Atomic Fallacy**: they treat knowledge as a collection of isolated, zero-dimensional points (isolated facts) for operational tractability, fundamentally misaligning with the high-dimensional inference paths that actually govern model reasoning (Mitchell et al., 2021; Jang et al., 2023; Yao et al., 2024; Chao et al., 2025). This structural discord exposes the inadequacy of atomic interventions, demanding a shift toward a graph-centric paradigm where trust and safety are secured by regulating relational connectivity rather than manipulating isolated points.

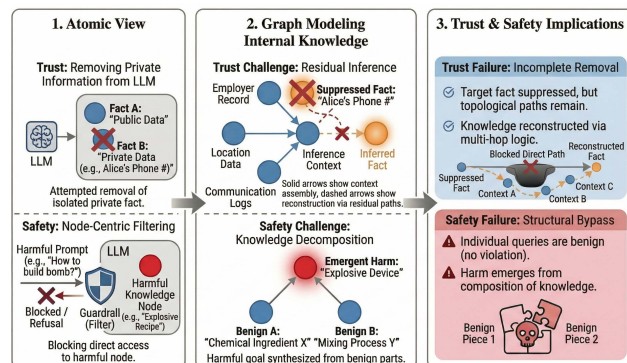

*Figure 4.* **Illustration of a Graph View of Internal Knowledge in LLMs for Trust and Safety.**

**The Structural Failure of Atomic Erasure.** A core challenge in LLM trust is the precise removal of sensitive, private, or copyrighted information, a task primarily addressed via machine unlearning. Current paradigms prioritize optimization techniques, such as gradient ascent (Jang et al., 2023), NegGrad+ (Kurmanji et al., 2023), or SCRUB (Kurmanji et al., 2023), to minimize the likelihood of specific target sequences. However, such fact-centric approaches frequently yield only superficial unlearning, as they neglect the model's underlying relational structure. As illustrated in Fig. 4, even if a specific fact (e.g., "Alice's phone number") is explicitly unlearned, the information often remains reconstructible through inferential links to associated entities, such as employer records or location history.

The failure of atomic erasure is rooted in the phenomenon of topological persistence. Empirical probing demonstrates that even when specific relation triples are technically forgotten by directly querying, the underlying information still survives in the knowledge network and can be reconstructed via multi-hop inference (Wu et al., 2024a). Indeed, analyses reveal that over 30% of target knowledge remains reachable through residual topological traces (Wei et al., 2025a) in models such as Llama-3 and Qwen-2.5, even after the application of state-of-the-art unlearning methods. Similar issues also mirror the "ripple effects" in knowledge editing, where isolated updates trigger global inconsistencies (Cohen et al., 2024; Shi et al., 2024). Although factual knowledge can be localized to specific weights (Meng et al., 2022a;b; Tan et al., 2023), manipulating these parameters in isolation tends to ignore the relational dependencies, leading to failures in compositional reasoning (Zhong et al., 2023).

**Fragmented Defenses Fail Against Structural Knowledge Exploitation.** In the realm of safety, this "Atomic Fallacy" manifests as a reliance on fragmented defense mechanisms. Instead of addressing structural vulnerabilities, current research largely pursues adversarial robustness through piecemeal red-teaming and alignment (Wang et al., 2025b), treating attack queries as isolated instances rather

than components of a broader strategy (Liu et al., 2023; Chao et al., 2025; Zeng et al., 2024). Consequently, deployed guardrails such as Llama Guard (Inan et al., 2023) and Qwen Guard (Zhao et al., 2025a) enforce policies on a strict per-prompt basis, ignoring the deeply correlated nature of LLM internal knowledge. This node-centric defenses leave models vulnerable to the compositional attacks illustrated in Fig. 4.

Prior work consistently demonstrates that decomposing harmful objectives into correlated, benign components is sufficient to bypass current safety mechanisms. This exploitation manifests across a spectrum of structural complexity. Adopting static input decomposition, attackers fragment intent into seemingly benign prompt structures. Static decomposition fragments intent into seemingly benign structures, achieving over 70% success in malicious code generation (Wahréus et al., 2025) and 80% in general jailbreaking (Srivastav & Zhang, 2025; Wei et al., 2023). Pushing this concept to its extreme, CKA-Agent (Wei et al., 2025b) actively leverages the target LLM as a structured knowledge guider to conduct dynamic and adaptive attack, achieving nearly 95% success even on robust frontier models such as Gemini 3.0, GPT-5.2, and Claude 4.5. Most recently, following a similar principle, Chain-of-Thought hijacking embeds harmful intent within globally unsafe but locally benign reasoning trajectories, effectively achieving safety violations (Zhao et al., 2025b). Collectively, these results indicate that while current defense mechanisms handle isolated prompts effectively, they remain vulnerable to attacks embedded within correlated reasoning structures.

**Call to Action: Towards Topological Defense.** The evidence presented underscores the urgent necessity of a paradigm shift toward graph-based modeling to combat knowledge leakage and adversarial bypass. To fully realize this vision of Topological Defense, several critical frontiers require immediate exploration. From an evaluation standpoint, there is an urgent need for domain-specific benchmarks equipped with structure-aware metrics. High-stakes domains, such as healthcare, finance, and law, require relational datasets capable of assessing how methods govern correlated knowledge, rather than merely checking atomic facts. Evaluation criteria must reflect domain-dependent risk tolerance: while highly sensitive content (e.g., PII or dual-use bio-threats) demands the severance of both direct queries and all plausible indirect inference paths, less restrictive settings (e.g., IP protection) may prohibit direct reproduction while tolerating limited conceptual relevance to preserve utility. Consequently, future frameworks should establish principled trust metrics calibrated to the domain-specific depth of topological dependency.

Given the growing success of decomposition-based attacks, we argue that defenses must evolve toward graph-based

multi-turn monitoring. Prior initiatives such as SentinelAgent (He et al., 2025) and GUARDIAN (Zhou et al., 2025a) have explored safety monitoring over explicit interaction graphs within multi-agent workflows (e.g., email scenarios). However, methodologies that track implicit knowledge dependencies to defend against internal structural vulnerabilities are still limited. A promising frontier lies in detecting malicious intent through the structural analysis of a user's trajectory. Approaches such as deliberative defense (Guan et al., 2024) attempt to conduct user's intent analysis by incorporating reasoning-based defense. However, CKA-Agent (Wei et al., 2025b) still reveals the fragility of the defense: single-session, dynamic decomposition attacks still hold success rates exceeding 80% to GPT-5.2 and GPT-OSS-120B even with deliberative defense. Finally, extending structural knowledge modeling to multimodal and multi-agent ecosystems presents a critical need, but this may require incorporating "Theory of Mind" to govern information propagation in collaborative environments, which introduces new and interesting challenges.

## 3. Alternative Views and Conclusion

**The Scaling Hypothesis.** The most prominent counter-argument to our position is rooted in the "Scale-First" hypothesis: the belief that relational and structural complexities are best handled by linearizing data into massive sequences and allowing general intelligence to emerge through sheer scale (Sutton, 2019). From this perspective, injecting explicit structural priors or graph constraints is viewed as a human-centric bias that may inadvertently limit the model's capacity to discover optimal latent representations.

**Our Response: Structural Priors as a Scaling Multiplier.** We do not propose Graph Learning as a replacement for scaling, but as a refinement that renders it more cost-effective and logically robust. We contend that GL confers critical advantages that scale alone struggles to guarantee. By anchoring generation to graph-structured memory, we increase retrieval precision for multi-hop queries and filter out semantic distractors, while enabling agents to maintain robust dynamic state tracking for long-horizon tasks. Graph-native architectures remove positional biases and computational redundancy inherent in naive serialization, just as mapping neural dynamics to discrete graphs transforms interpretability from post-hoc rationalization into verifiable logic. Finally, recognizing that safety is fragile under structural bypasses (e.g., decomposition-based attacks), we argue that the topological governance of LLMs' knowledge dependencies is crucial for next-generation defense. Ultimately, we advocate that the field must move beyond a singular focus on scaling, treating graph-based topological dependencies as a necessary scaffolding for the next generation of trustworthy foundation models.

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
