# OpenReview forum: "Position: Beyond the Sequence -- Graph Learning as the Blueprint for Trustworthy Large Language Models"
_ICML.cc/2026/Position_Paper_Track — Submitted to ICML 2026 Position Paper Track_

### Official Review · Reviewer_EqKX · 2026-03-08

**Significance:** 3
**Argument Clarity:** 3
**Rating:** 5
**Confidence:** 4

**Questions:**

Comments / Questions

- A recent position/survey paper that the authors should cite and might also want to take inspiration from is (Arroyo et al. 2025), which argues that many of the failure modes observed in transformer-based language models are a consequence of some of the information propagation issues in graph neural networks, including oversmoothing and over-squashing. This is a complementary view/position that the authors might want to consider.

Arroyo, Álvaro, et al. "Bridging Graph Neural Networks and Large Language Models: A Survey and Unified Perspective." (2025).

**Alternative Views Section:**

Yes

**Compliance With Llm Reviewing Policy A Conservative:**

Affirmed.

**Discussion Potential:**

3

**Final Justification:**

I believe that the paper is well executed and touches upon a timely and interesting topic. With their response, the authors have addressed most of my comments / concerns, which has made me raise my score.

**Paper Summary:**

This paper proposes different ways through which principles from graphs and networks can be applied in the context of Large Language Models (LLMs). The authors place a focus on (a) Context engineering (b) Graph Priors on Architectures (c) Explainability through Graphs (d) LLM Safety.

**Position:**

Yes

**Position In Title:**

Yes

**Related Work:**

3

**Strengths And Weaknesses:**

Pros

- I think that the idea of using graph-based ideas to understand and improve language models is novel and interesting, and not too explored in the literature, which makes this a good position paper to introduce people to the topic at hand.
- The paper is presented in a compelling way and has nice figures to illustrate concepts.

Cons

- In Section 2.1, there is just a mention of the advantage of GraphRAG when considering distractors. The authors talk about the possibility of having an evolving graph memory, but do not really propose any mechanism by which this should function. Most of the arguments in these sections are presented without proof or examples, which makes it more like a wishlist rather than proof that these proposals could work in practice.
- The latter parts on the use of LLMs also seem similarly weak in this sense, where concepts are discussed with the modus operandi mainly being that of presenting ideas and interleaving them with some of the related literature, but without showing very precise examples of how graph learning is useful in these settings. While this is a bit vague, I would recommend that the authors include small examples of how incorporating graph learning-based inductive biases is helpful for the subproblems mentioned. These do not necessarily need to be novel, but would be useful to illustrate the main points the authors are trying to convey.
- I also find the paper somewhat lacking in terms of the mathematical formalism of the concepts presented. I think that this would allow the authors to ground the concepts more clearly beyond descriptions and mentions of past literature.

**Support:**

2

---

> ### Author Rebuttal · Authors · 2026-03-25
>
> ## W1. Missing mechanism for evolving graph memory
>
> We thank the reviewer for this constructive feedback and agree that the mechanics of evolving graph memory can be made more prominent.
>
> Our intent in Sec. 2.1 was to outline a concrete design space: nodes represent entities or episodic states, edges encode causal or relational dependencies, and resolved subtasks can be folded into supernodes while preserving pointers for later unfolding. We also discuss an optimization pathway in which graph operations such as parsing, folding, and linking are treated as learnable actions that can be optimized end-to-end through RL or process rewards. As a position paper, **our aim is not to claim that a single canonical mechanism has already been established, but rather to articulate a plausible and increasingly tractable direction.**
>
>
> In addition, after submission, we became aware of very recent work, MemEvolve [1], which further supports this direction by showing that memory architectures themselves can be optimized via evolutionary search and can yield meaningful gains on agentic benchmarks.
>
> We will ensure these concrete mechanisms are highlighted more clearly in the final manuscript.
>
> [1] MemEvolve: Meta-Evolution of Agent Memory Systems, 2025.
>
> ## W2. Missing Concrete Examples in Secs. 2.2–2.4
>
> We appreciate this suggestion and agree that the connection between the paper’s conceptual claims and the concrete examples can be made much clearer. Our intention was for the figures and examples in these sections to play exactly this role.
>
> - **Sec 2.2 (Architecture)**: We highlight BlockRank and Graph-KV as concrete implementations of structure-aware architectures. Specifically, we detail how Graph-KV enforces a strict topological mask (DAG-masked attention) combined with graph-aligned positional encoding. As shown in Fig. 2, evaluating this on the ARXIV-QA citation network demonstrates that it successfully reduces computational overhead while maintaining high accuracy regardless of context serialization order.
>
> - **Section 2.3 (Explainability)**: Fig. 3 provides a specific walkthrough using the reasoning trace for "2*4+2". It concretely demonstrates how mapping neural dynamics to implicit graphs exposes internal causal circuits, and how explicit graphs allow for targeted interventions, such as pruning invalid reasoning steps to enforce structural consistency. We explicitly anchor this discussion in established mechanistic frameworks such as ACDC, EAP, and CRV.
>
> - **Section 2.4 (Safety)**: Fig. 4 visualizes concrete structural safety failures. For unlearning, we detail the "Atomic Fallacy": even if an isolated fact like "Alice's phone number" is effectively suppressed, it remains vulnerable to reconstruction via correlated nodes, such as employer or location records. We cite empirical evidence showing that over 30% of target knowledge persists through these residual topological traces. For jailbreaking, we discuss how harmful intents are decomposed into individually benign prompts —a structural exploit that achieves nearly 95% success rates against frontier models. Together, these examples underscore the necessity of graph-aware safety analysis.
>
> We agree, however, that the linkage between our conceptual arguments and these specific empirical examples can be highlighted  more clearly in the revised version.
>
> ## W3. Lack of math formalism
>
> We appreciate the reviewer's feedback and agree that the paper could benefit from greater mathematical precision.
>
> Because this is a position paper, our intent was to provide a lightweight formal grounding to outline a design space, rather than a full theoretical treatment. In fact, the current draft does establish some formalization. For instance, in Sec. 2.2, we represent context states using graph notation, and also explicitly define graph-constrained attention in terms of ancestor relations over a DAG. Furthermore, our arguments rely on rigorous graph-theoretic abstractions throughout the text, including message passing and permutation symmetry.
>
> ## The suggested literature
>
> We thank the reviewer for pointing out Arroyo et al. (2025). We discovered this concurrent work shortly after submission and agree it is a highly relevant, complementary reference for our discussion on architectural inductive biases. We will explicitly cite and discuss it in Sec. 2.2.
>
> Arroyo et al. provide a unified graph perspective on decoder-only Transformers, linking GNN phenomena—like over-smoothing and over-squashing—to Transformer information-flow bottlenecks. While their theoretical analysis perfectly complements Sec. 2.2, their primary focus is on the mechanics of Transformer behavior. In contrast, our position paper takes a broader, system-level view of integrating graph methodologies across the entire LLM lifecycle, spanning context engineering, architecture, interpretability, and safety. Incorporating their insights will strengthen our architectural arguments.

---

> > ### Author Rebuttal · Reviewer_EqKX · 2026-04-02
> >
> > I thank the authors for their response. I believe that t clarified my questions / points, and I will take it into account in my final score decision.

---

### Official Review · Reviewer_JTMc · 2026-03-11

**Significance:** 3
**Argument Clarity:** 2
**Rating:** 4
**Confidence:** 3

**Questions:**

- Figure 1 shows (i) and (ii) but the caption states a) and b), it is unclear what is meant here.

- Following the weakness section, while the experiments provide some insight and support for the arguments, more recent baselines and more comprehensive empirical evidence would be beneficial.

**Alternative Views Section:**

Yes

**Compliance With Llm Reviewing Policy A Conservative:**

Affirmed.

**Discussion Potential:**

3

**Final Justification:**

My final recommendation is borderline accept. The authors' rebuttal has mainly addressed my concerns and I look forward to seeing better revision of the paper in the areas that I mentioned.

**Paper Summary:**

Modern deep learning often assumes that scaling on massive datasets is enough to achieve general intelligence. This paper's position is that  graph learning serve as a structure multiplier to the statistical power of scaling. The authors identify a core problem: recovering relational topology from sequences is resource-intensive and leaves gaps in factual grounding, logic, and transparency. They propose four integration points between GL and LLMs including context engineering, architecture, interpretability, and safety, framing GL not as a competing approach but as structural scaffolding that makes scaling more efficient and its outputs more verifiable.

**Position:**

Yes

**Position In Title:**

Yes

**Related Work:**

2

**Strengths And Weaknesses:**

Overall, the paper points out various directions to incorporate graph learning and the existing LLM pipelines. The position is relatively clear though it is also very broad as there are many ways graph and sequences can be utilised together in foundation models.

Strengths:

- **S1** Graph structure is mostly overlooked in the current LLM design, though has been recent interest and discussion in this direction which aligns with the position of this paper. This paper can help facilitate more discussion in this direction.

- **S2** The position presented by the paper is clear and the authors discussed four directions to incorporate graph learning with LLM.

- **S3** The paper is well-written and easy to follow.


Weaknesses:
-  **W1** While using the topology might boost performance, how to construct the graph and retrieve the relevant graph is a difficult challenge to begin with. The paper should discuss more on how to structure the graph.

- **W2** The paper also feels disjointed, and there is no conclusion section. The paper covers four distinct areas in the LLM research including contexting engineering (2.1), priors of architectures (2.2),  explaining LLMs (2.3) and LLM safety (2.4). These topics utilizes the graph structure in different ways and overall makes the papers more similar to a survey about graph and LLMs rather than a coherent position paper that argues over a centralized point. The paper can benefit from zooming into one of two of these topics and clearly demonstrate the advantage of graph learning with stronger empirical evidence.

- **W3** Table 1 experimental setting is unclear, what is the baseline that doesn't use PPR (in Table 1 a). Also the baselines used in Table 1 b). is mostly out-dated in the LLM literature, no longer reflecting the state-of-the-art in LLM reasoning power.

**Support:**

2

---

> ### Author Rebuttal · Authors · 2026-03-25
>
> **Thank you for the constructive feedback and for recognizing the value of our work.**
>
> We address your specific concerns below:
>
> ## W1: Graph construction and retrieval are difficult challenges.
> We agree that graph construction is a major practical hurdle, particularly for context engineering (Sec. 2.1) and architecture (Sec. 2.2). We explicitly acknowledge this bottleneck in Sec. 2.1, noting that optimal graph granularity is highly task-dependent. We will revise the manuscript to highlight this challenge more prominently and clarify why these directions are increasingly tractable:
>
> ### For Sec. 2.1 and Sec. 2.2 (External Graphs), we will highlight several promising directions that mitigate this concern:
>
> - **A Shift to End-to-End Optimization**: We explicitly argue that graph construction should not remain a fixed, manual preprocessing step. Instead, we advocate for end-to-end structural optimization, where parsing, folding, and linking operations become optimizable components of the reasoning trace. Recent work like MemEvolve [1] supports this by showing that memory architectures can be meta-evolved rather than statically designed.
>
> - **Proven Utility in Agentic Workflows**: Systems like OpenSage [2] and Tongyi DeepResearch [3] already demonstrate the practical viability and necessity of graph structured workflows for long-horizon reasoning and complex state tracking. Graphs here naturally come from agent workflows.
>
> ### By contrast, the role of graph structure shifts in Sec. 2.3 and Sec. 2.4:
>
> - **Modeling Internal Computation, Not External Retrieval**: In these domains, the graph is not an external object that must be built and searched. Rather, it is a framework to model the implicit internal computation and knowledge structures already present within the LLM. For interpretability, this is grounded in mechanistic work like ACDC and EAP, which uncover sparse component circuits to track how specific concepts propagate. For safety, graph-based modeling is crucial because vulnerabilities often bypass prompt filters by emerging through correlated relational paths in the model's internal knowledge.
>
> Overall, while external graph construction (Sec. 2.1 & 2.2) remains a challenge, emerging frameworks as above offer mitigation paths. Meanwhile, our graph-centric proposal in Sec. 2.3 & 2.4 rely on internal structures, rendering them independent of external graph retrieval constraints.
>
> [1] MemEvolve: Meta-Evolution of Agent Memory Systems, 2025.
>
> [2] OpenSage: Self-programming Agent Generation Engine, 2026.
>
> [3] Tongyi DeepResearch Technical Report, 2025.
>
>
> ## W2: The paper feels disjointed rather than a coherent position paper.
>
>
> We appreciate this feedback. The breadth of the paper is intentional. Our reason for structuring the paper this way is that graph usage in LLMs can play different functional roles across the lifecycle: organizing context (Sec. 2.1), providing architectural priors for computation (Sec. 2.2), exposing internal reasoning and mechanism structure (Sec. 2.3), and modeling connected knowledge and risk propagation for safety and unlearning (Sec. 2.4). In other words, the pillars are meant to reflect different intentions of graph usage in LLMs, rather than a loose collection of graph-related works.
>
> **Moving Beyond GraphRAG**: A primary motivation for structuring the paper this way is that current community discourse is mostly on engineering-oriented applications like GraphRAG, while other, potentially more fundamental roles of graph structure remain underexplored. In particular, broader perspectives such as **DAG-structured agent workflows**, **topological alignment for interpretability**, and **graph-centric views of unlearning and jailbreak propagation** are emerging in recent literature, but have not yet been clearly articulated as part of a shared graph-learning agenda for LLMs.
>
> In this sense, we aim to provide the community with a more holistic view of where graph learning may matter, and to open up additional angles for future research. We agree, however, that this unifying position should be stated more clearly. In the revision, we will strengthen the introduction and section transitions so that the manuscript reads more like a coherent argument about the broader structural role of graph learning in LLMs.
>
>
> ## W3: Table 1 clarity and empirical baselines.
>
>
> - Table 1(a): The non-graph baseline is BM25. This comparison strictly isolates the performance gain achieved by adding graph-enhanced retrieval (BM25 + PPR).
>
> - Table 1(b): This is a targeted ablation, not a general SOTA benchmark. It is designed solely to test distractor robustness and demonstrate how graph-organized context reduces semantic interference. However, we agree that evaluating newer architectures adds value. Please check [the linked table](https://storage.googleapis.com/review-materials/fig1.png).
>
> - Figure 1: Thank you for catching it. We will correct the labeling mismatch in the revision.

---

> > ### Author Rebuttal · Reviewer_JTMc · 2026-04-03
> >
> > Thank you for the detailed rebuttal and addressing my concerns. Looking forward to seeing the changes from the paper revision as well. I have raised my score accordingly.

---

### Official Review · Reviewer_dPve · 2026-03-15

**Significance:** 3
**Argument Clarity:** 3
**Rating:** 4
**Confidence:** 3

**Questions:**

I'd love to see some evidence that compares graph-based and sequential-based methods on the similar model size, same training data/compute.

**Alternative Views Section:**

Yes

**Compliance With Llm Reviewing Policy A Conservative:**

Affirmed.

**Discussion Potential:**

3

**Paper Summary:**

This paper says that sequence-first scaling is not sufficient for building trustworthy LLMs, and that graph learning should be treated as a complement to scale rather than as an optional add-on. The central problem discussed by the study is the mismatch between essentially related real-world structure and the one-dimensional sequential token representation used by standard LLMs. The author explores whether graph-based priors can improve grounding, reasoning, interpretability, and safety more effectively. The paper forms its position around four blueprints: graph-based context engineering, graph-informed architecture, graph-based interpretability, and graph-centric safety. It supports these ideas with conceptual arguments, and a few empirical examples.

**Position:**

Yes

**Position In Title:**

Yes

**Related Work:**

3

**Strengths And Weaknesses:**

Strength:

1. The paper is ambitious and timely. The core challenge that that relational structure is inefficiently rediscovered by sequential models and should sometimes be injected explicitly is intuitive and relevant to several active areas. Framing GL as a scaling multiplier rather than a replacement for scaling is also a smart choice that makes the position more balanced and plausible.

2. The four-blueprint structure gives the manuscript a backbone: graph-structured memory for context engineering, graph priors for attention, graph abstractions for interpretability, and graph-based knowledge connectivity for safety and unlearning. This makes the position more like a research discussion than a loose collection of claims.

Weakness:

1. The paper's coverage is too broad relative to the evidence provided. It makes strong claims over retrieval, architecture, interpretability, unlearning, etc, but the levels of empirical support vary a lot across sections. Context engineering and attention get some concrete evidence. Interpretability and safety are much more speculative.

2. Many of the individual ideas discussed here are already active lines of work: GraphRAG, graph memories for agents, sparse/topological attention, graph-based interpretability, and graph views of unlearning. I feel the paper’s contribution is mainly on synthesis and positioning.

3. The paper sometimes overstates against sequential models. The argument that serialization causes persistent deficits in factual grounding, logic, and transparency may be directionally true, but it is not obvious that these are problems that cannot be significantly reduced by scale, better objectives, tool use, etc.

**Support:**

3

---

> ### Author Rebuttal · Authors · 2026-03-25
>
> Thank you for your encouraging review. We appreciate your recognition of our work as “ambitious and timely.”
>
> ## Regarding clearly apples-to-apples evaluations
>
> We fully agree that controlled comparisons are important. Our intent is not to claim that graph-based methods have already been shown to outperform sequential methods across all four pillars. Rather, our claim is narrower: existing evidence suggests that graph priors are promising complements that merit more systematic evaluation. In Sec. 2.1, graph retrieval is evaluated in a relatively controlled setting by adding PPR reranking on top of the same BM25 base retriever, which improves retrieval recall and also improves long-context QA F1 for the same downstream LLMs. In Sec. 2.2, all models in Fig. 2 are variants of Llama 3.1-8B, and Graph-KV is substantially more robust to serialization order than sequential and block-based baselines.
>
> By contrast, Secs. 2.3 and 2.4 are not primarily about end-task model performance, so they are less naturally framed through direct apples-to-apples comparisons. We address their evidence below in response to W1.
>
> ## W1: The paper is broad relative to the level of evidence.
>
> We agree that the paper is broad, and this is intentional. Current discussion of graphs for LLMs is still concentrated mainly around RAG and engineering practice. Our goal is to take a broader lifecycle view of where graph learning may be beneficial.
>
> In fact, we argue that the evidence for Interpretability and Safety has been provided in the paper:
>
> - **Evidence in Safety (Sec. 2.4)**: Even when specific relation triples are unlearned and forgotten under direct querying, the underlying information often survives in LLMs and remains reconstructible via multi-hop inference. As discussed in the paper, recent probing analyses demonstrate that over 30% of target knowledge remains reachable through residual topological traces after applying unlearning methods. Furthermore, current guardrails fail because harmful objectives can be decomposed into correlated and structured benign components, which may achieve over 90% attack success on robust frontier LLMs. These findings suggest that the underlying vulnerabilities are structural, which motivates our graph-centric view of defense.
>
> - **Evidence in Interpretability (Sec. 2.3)**: The interpretability section is similarly grounded in empirical evidence. Frameworks such as ACDC and EAP have uncovered sparse component circuits within dense models. Recent research further recovers fine-grained information-flow graphs that reveal how specific concepts propagate throughout the network. These graph abstractions also enable active intervention and verification: for example, CRV uses graph-level structural properties to distinguish reliable reasoning from hallucinations.
>
> ## W2: Many of the ideas are already active lines of work.
>
> We agree that some domains, especially GraphRAG, are already active areas. However, we contend that from a graph-modeling perspective, **the other three pillars—architecture, interpretability, and safety—remain comparatively less developed and have not yet been clearly organized into shared research directions**. Highlighting and organizing these directions is the main gap this position paper aims to fill.
>
> Even within established areas like RAG and agentic memory, our paper highlights some underappreciated perspectives:
>
> - GraphRAG as a Noise Filter (Sec 2.1): Beyond multi-hop retrieval, graph-organized retrieval actively mitigates semantic interference. As Table 1 shows, it reduces harmful semantic distractors to provide a "better noise profile". Graph structures fundamentally shape a cleaner context window for robust LLM reasoning.
>
> - Beyond Static Graph Databases for Agents: While basic graph backends appear in agents, we articulate a graph-native view of dynamic state tracking. We propose trackable, foldable, and unfoldable memory structures, a largely untapped frontier for long-horizon agentic workflows.
>
> ## W3: Overstating the case against sequential models.
>
> We appreciate this point and apologize if the current writing sounds overly absolute. Our intention is not to argue that sequential models cannot be substantially improved by scaling or better data curation. On the contrary, we recognize and appreciate the impressive progress made along these directions.
>
> Our intended claim is narrower: graph-based priors may provide a complementary and sometimes more competitive alternative in settings where explicit relational structure matters. In this sense, the paper is not arguing against progress in sequence-first modeling, but rather advocating that the field should think more broadly about trustworthy LLM design beyond scaling alone.
>
> We will revise the manuscript to explicitly soften our assertions against sequence-only models. We thank the reviewer for highlighting this point, as addressing it will improve the balance and maturity of the paper.

---

> > ### Author Rebuttal · Reviewer_dPve · 2026-04-05
> >
> > I appreciate the response from author that addressed my concerns on wording. I'm willing to keep positive view for now.

---

### Official Review · Reviewer_V8FB · 2026-03-16

**Significance:** 4
**Argument Clarity:** 3
**Rating:** 5
**Confidence:** 4

**Questions:**

While self-attention scales quadratically with the number of input tokens, it adapts well to the computational architectures of GPUs. I wonder if graph structures embedded in the architecture, even if they could, in principle, introduce sparsity, would ultimately result in slower inference speed.

**Alternative Views Section:**

Yes

**Compliance With Llm Reviewing Policy A Conservative:**

Affirmed.

**Discussion Potential:**

4

**Final Justification:**

I believe this work would be an interesting read for researchers in the fields of bot graph learning and LLM research.  My original positive recommendation was confirmed by the authors' answer to my initial minor concerns.

**Paper Summary:**

The paper position is that applying graph learning to LLM architectures would be beneficial in many contexts. In particular the paer identifies four areas of intervention:

- Graph-based context engineering: the flat sequence operational mode of current LLMs is unable to properly handle the inherent hierarchical structure of documents in a RAG scenario and agentic memory. A graph-based structuring of the information would benefit the model in handling hierarchical and traversal relations.

- Graph architectural priors: this would allow the model to organize its internal memory and processing in a structured, hierarchical way, better mimicking the reasoning process. In particular, the paper advocates for a Directed Acyclic Graph structure.

- Explainability: the dense attention approach of LLMs does not allow for applying ad-hoc interpretability strategies, often resulting in post-hoc explanations that poorly reflect the internal decision process of the model. The paper suggests that if the causal reasoning of the model is made explicit through a graph-like structure of the architecture, this would help both interpretability and intervention on the model.

- The last intervention area regards LLMs safety and, in particular, machine unlearning. The literature states that the standard query-based unlearning of specific atomic information does not prevent that information from being retrieved from alternative indirect paths. The paper calls for both a graph-based defence and an evaluation of such attacks that account for the data's topological structure.

**Position:**

Yes

**Position In Title:**

Yes

**Related Work:**

4

**Strengths And Weaknesses:**

The paper is quite dense, but well-organized in four intervention areas. Each point is well-grounded in the literature, with an overview of what has been done so far and what the open challenges are according to the authors. I think that the paper would be an interesting read for both the graph and the LLM community.

The paper is missing an actual conclusion that summarizes the content of the paper.

The paper does not propose a completely new approach, but rather it makes order on the current state of GL applied to LLMs, identifying 4 main intervention areas. Even if this is still a valuable contribution, since the paper is quite dense, it is not always easy to discern the authors’ novel ideas from the approaches already present in litterature. Making this aspect more explicit could help the reader.

**Support:**

4

---

> ### Author Rebuttal · Authors · 2026-03-25
>
> Thank you for your encouraging review. We appreciate you recognizing the value of our four pillars. Currently, graph-LLM interaction is often discussed narrowly through the lens of RAG. Our core motivation is to broaden this perspective, framing graph learning as a structural multiplier across the entire LLM lifecycle.
>
>
> ## W1: Clarification on Novel Perspectives
>
>
> While our supporting evidence draws from recent literature, our intended contributions lie in unifying these scattered, implicit and unclear signals into a cohesive framework where graph learning serves as a foundational structural layer for LLMs. We aim to move the field's conversation on the potential of graph learning in modern AI systems beyond merely GraphRAG:
>
> - **Context engineering**. We highlight an underappreciated implication of recent empirical results: graph-organized context can reduce harmful distractors by mitigating semantic interference. This benefit remains insufficiently recognized in much of the current GraphRAG community, yet it could significantly enhance dynamic agentic memory.
>
> - **Architecture**: We emphasize that agent workflows naturally form DAGs of tool usage and state dependencies. This suggests that graph structures can be naturally integrated into the sequential generative process of LLMs, advancing beyond earlier simplified forms like independent-set and block-sparse attention.
>
> - **Interpretability**. Topological alignment between the explicit reasoning graph and the implicit computation graph is needed. This ensures that graph-based explanations are sufficiently faithful to support actual intervention, verification, regulation, and mechanism monitoring.
>
> - **Safety**. We raise the "Atomic Fallacy," arguing that the real risk lies not only in isolated facts but in correlated, topologically connected knowledge paths. This calls for topological defense and structure-aware evaluation rather than purely atomic interventions.
>
>
> ## W2: GPU efficiency / practicality of graph-structured computation.
>
>
> We agree that GPU efficiency is a valid systems concern (for Section 2.2). We do not claim graph priors automatically deliver faster wall-clock inference in every implementation. However, this issue is practically mitigated in two ways, which we will explicitly clarify in the revision:
>
> - **Staged Adoption**. Large-scale pretraining can remain fully sequential and dense. Graph-structured sparsity is introduced primarily during alignment or fine-tuning. Methods like BlockRank and Graph-KV already successfully follow this pattern by applying structural priors to off-the-shelf LLMs.
>
> - **Coarse & Regularized Sparsity**: We advocate for block-level, regular sparsity rather than arbitrary, fine-grained computation. For example, Graph-KV operates over regular-length text blocks with DAG-restricted attention and graph-aligned positional encoding, which preserves much of the hardware parallelism inherent in blockwise computation.
>
> We will revise the manuscript to clarify this point explicitly: While naive graph sparsity may indeed be hardware-unfriendly, controlled and regularized graph-structured sparsity mitigates this concern while still offering highly useful structural biases.

---

> > ### Author Rebuttal · Reviewer_V8FB · 2026-04-03
> >
> > I thank the authors for their clarifications and appreciate the revisions they intend to incorporate into the manuscript. I have no further questions.

---

### Decision · Program_Chairs · 2026-04-30

**Decision:**

Reject

**Comment:**

The paper argues for the importance of graph structured processing, which would have multiple benefits compared to the sequential processing. The reviewers seem to agree of the value of the perspective. Overall I agree with the proposed position, however I think the position is widely acknowledge, for example through the graph network community. Overall while the position has some merit, given the high bard required by the conference, I think the position paper needs to emphasize more how this view is being ignored by the community.